# Robust Four-Wave Mixing and Double Second-Order Optomechanically Induced Transparency Sideband in a Hybrid Optomechanical System

Huajun Chen

School of Mechanics and Photoelectric Physics, Anhui University of Science and Technology, Huainan 232001, China; hjchen@aust.edu.cn

**Abstract:** We theoretically research the four-wave mixing (FWM) and second-order sideband generation (SSG) in a hybrid optomechanical system under the condition of pump on-resonance and pump off-resonance, where an optomechanical resonator is coupled to another nanomechanical resonator (NR) via Coulomb interaction. Using the standard quantum optics method and input–output theory, we obtain the analytical solution of the FWM and SSG with strict derivation. According to the numerical simulations, we find that the FWM can be controlled via regulating the coupling strength and the frequency difference of the two NRs under different detuning, which also gives a means to determine the coupling strength of the two NRs. Furthermore, the SSG is sensitive to the detuning, which shows double second-order optomechanically induced transparency (OMIT) sidebands via controlling the coupling strength and frequencies of the resonators. Our investigation may increase the comprehension of nonlinear phenomena in hybrid optomechanics systems.

**Keywords:** hybrid optomechanical systems; four-wave mixing; second-order sideband generation





## 1. Introduction

Cavity optomechanics (COM) systems [1,2] (as a milestone [3] in optics history), which investigates the interaction of electromagnetic fields and micromechanical motion, have witnessed significant progress over the past decade both in fundamental studies and practical applications including ground state cooling [4–8], mass sensing [9,10], high-precision measurements [11–17], and quantum information processing [18–21]. The mechanical motions in COM systems, due to the radiation pressure forces, are tunable by optomechanical interactions, which in turn influence the optical medes resulting in prominent quantum interference effects. There are many famous phenomena that have been obtained in COM systems, such as phonon lasers [22–25], squeezing [26,27], entanglement [20,21], nonreciprocity [28–30], exceptional point (EP) devices [24,25,31,32], optomechanically induced transparency (OMIT) [33–40], and OMIT induced slow and fast light [36,41–44]. We notice that many above phenomena remain in the linear optical regime.

COM systems also present a medium to research the nonlinear phenomena between the electromagnetic field light and matter. Optical bistability [45–49], as a representative nonlinear phenomena, has been extensively investigated in some kinds of COM systems. In the COM systems, if the cavity is driven by a strong pump laser field (with frequency $\omega_p$) and a weak probe laser field (with frequency $\omega_s$), then when the two pump photons mix with a probe photon, an idler photon at frequency $2\omega_p - \omega_s$ will emerge, as a result the four-wave mixing (FWM) appear in the ouput field, which has been investigated in different optomechanical systems [50–53]. Except nonlinear phenomena of optical bistability and FWM, recently, another remarkable nonlinear optomechanical effect, i.e., the second-order sideband generation (SSG), has also been demonstrated in COM systems [54–64], where the SSG will appear in the output fields with frequencies $\omega_p \pm 2\delta$ ($\delta = \omega_s - \omega_p$ is the probe-pump detuning) and $\omega_p + 2\delta$ ($\omega_p - 2\delta$) is the second-order upper (lower) sideband

frequency component. However, the FWM and SGG in a hybrid optomechanical system, where a typical optomechanical cavity coupled to another NR via Coulomb interaction has not been demonstrated until now.

In this paper, we investigate the FWM and SSG in a hybrid optomechanical system as shown in Figure 1 under the condition of pump on-resonance and off-resonance. The location of sideband peaks both in the FWM spectrum and SSG depends on the resonator frequencies and the Coulomb interaction of the two NRs. In particular, the different frequencies of the resonators also alter the location of the peaks in the FWM spectrum and SSG. Interestingly, in the pump off-resonance regime, the FWM spectrum can indicate a means to measure the Coulomb interaction, and we obtain double second-order OMIT sideband, which is sensitive to the Coulomb interaction and different resonator frequencies.

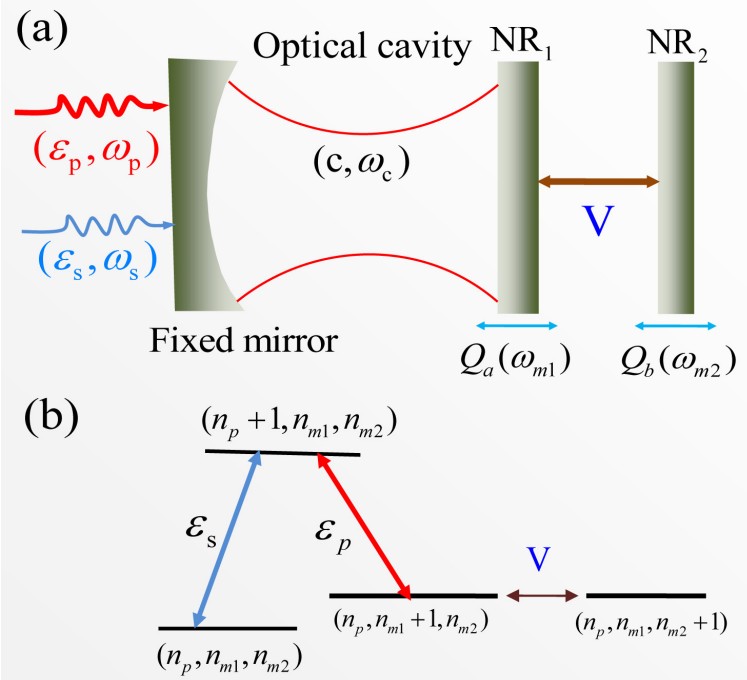

**Figure 1.** (**a**) Schematic diagram of the hybrid coupled optomechanical system including two coupled nanomechanical resonators (NRs) via Coulomb interaction. (**b**) Schematic of the energy-level diagram of the system. $|n_p\rangle$, $|n_{m1}\rangle$, and $|n_{m2}\rangle$ denote the number states of the cavity photon, the phonons of NR$_1$ and NR$_2$, respectively. $V$ is the coupling strength of the two NRs.

## 2. System and Method

The hybrid optomechanical system is shown in Figure 1, which includes a Fabry–Perot (FP) optomechanical cavity coupling to another NR via Coulomb interaction, and the Hamiltonian can be given by [65–67]

$$
\begin{aligned}
H =\ & \hbar\omega_c c^\dagger c + \hbar\omega_{m1} b_1^\dagger b_1 + \hbar\omega_{m2} b_2^\dagger b_2 - \hbar g c^\dagger c (b_1^\dagger + b_1) - \hbar V (b_1^\dagger b_2 + b_1 b_2^\dagger) \\
& + i\hbar\sqrt{\kappa_{ex}}\varepsilon_p (c^\dagger e^{-i\omega_p t} - c e^{i\omega_p t}) + i\hbar\sqrt{\kappa_{ex}}\varepsilon_s (c^\dagger e^{-i\omega_s t} - c e^{i\omega_s t}).
\end{aligned}
\tag{1}
$$

where the first term gives the cavity field (with frequency $\omega_c$) and we use the creation (annihilation) operator $c^\dagger(c)$ to describe the optical cavity. The second and third terms shows two NRs with the frequencies $\omega_{m1}$ and $\omega_{m2}$, $b_1^\dagger(b_2^\dagger)$ and $b_1(b_2)$ are the creation and annihilation operators of the two NRs, respectively. The fourth term gives the optomechanical coupling with coupling strength $g = \frac{\omega_c}{L}\sqrt{\frac{\hbar}{2M_1\omega_{m1}}}$, where $M_1$ is the effective mass of NR$_1$ and $L$ is the cavity length. The fifth term describes the interaction of two charged NRs with coupling strength $V$ via the Coulomb interaction [65–67]. As the cavity is driven by a two-tone fields, we define $\varepsilon_p = \sqrt{P_c/\hbar\omega_p}$ and $\varepsilon_s = \sqrt{P_s/\hbar\omega_s}$ as the amplitudes of the

two laser fields, and $P_c$ and $P_s$ are their powers. $\kappa_{ex}$ is extra loss rate and here we consider $\kappa_{ex} = \kappa_0$, where $\kappa_0$ is the intrinsic loss rate of photon with the relation of $\kappa = \kappa_{ex} + \kappa_0$.

We can rewrite Equation (1) in the following in a frame rotating with the frequency $\omega_p$ [65–67]

$$
\begin{aligned}
H &= \hbar\Delta_c c^\dagger c + \hbar\omega_{m1}b_1^\dagger b_1 + \hbar\omega_{m2}b_2^\dagger b_2 - \hbar g c^\dagger c(b_1^\dagger + b_1) - \hbar V(b_1^\dagger b_2 + b_1 b_2^\dagger) \\
&\quad + i\hbar\sqrt{\kappa_{ex}}\varepsilon_p(c^\dagger - c) + i\hbar\sqrt{\kappa_{ex}}\varepsilon_s(c^\dagger e^{-i\delta t} - c e^{i\delta t}).
\end{aligned}
\tag{2}
$$

where $\Delta_c = \omega_c - \omega_p$ is the cavity-pump detuning. We then obtain the Langevin equations (LEs) as follows with adding the corresponding damping and input noise terms [33,35,65–67]

$$
\dot{c} = -(i\Delta_c + \kappa)c + igcQ_a + \sqrt{\kappa_{ex}}(\varepsilon_p + \varepsilon_s e^{-i\delta t}) + c_{in},
\tag{3}
$$

$$
\ddot{Q}_a + \gamma_{m1}\dot{Q}_a + (\omega_{m1}^2 + V^2)Q_a + V(\omega_{m1} + \omega_{m2})Q_b = 2g\omega_{m1}c^\dagger c + \xi_1,
\tag{4}
$$

$$
\ddot{Q}_b + \gamma_{m2}\dot{Q}_b + (\omega_{m2}^2 + V^2)Q_b + V(\omega_{m1} + \omega_{m2})Q_a = 2gVc^\dagger c + \xi_2,
\tag{5}
$$

where the input vacuum noise is denoted by $c_{in}$ with zero mean value, and $\xi_1$ and $\xi_2$ are Langevin force arising from the environment. $\gamma_{m1}$ and $\gamma_{m2}$ are the decay rates of the two NRs. $Q_a = b_1^\dagger + b_1$ and $Q_b = b_2^\dagger + b_2$ are the position operators of the two NRs.

Due to the pump field being stronger than the probe field, we use the conversion of $O = O_s + \delta O$ ($O = c, Q_a, Q_b$), i.e., the operators are divided into the steady-state mean value and a small fluctuation. For the steady-state values, which is determined by the following equations

$$
(i\Delta + \kappa)c_s = \sqrt{\kappa_{ex}}\varepsilon_p,
\tag{6}
$$

$$
\omega_{m1}'^2 Q_{as} + V' Q_{bs} = 2g\omega_{m1}|c_s|^2,
\tag{7}
$$

$$
\omega_{m2}'^2 Q_{bs} + V' Q_{as} = 2gV|c_s|^2,
\tag{8}
$$

where $\Delta = \Delta_c - gQ_{as}$, $\omega_{m1}'^2 = \omega_{m1}^2 + V^2$, $\omega_{m2}'^2 = \omega_{m2}^2 + V^2$, and $V' = V(\omega_{m1} + \omega_{m2})$. In the condition of mean-field approximation, the operators can be replaced by their expectation values $\langle xc \rangle = \langle x \rangle \langle c \rangle$ [33], For the fluctuation operators, we also use their expectation values with neglecting nonlinear terms, and we obtain the expectation values of the LEs [33]

$$
\langle \delta\dot{c} \rangle = -(i\Delta + \kappa_c)\langle \delta c \rangle + igc_s\langle \delta Q_a \rangle + ig\langle \delta c \rangle\langle \delta Q_a \rangle + \sqrt{\kappa_{ex}}\varepsilon_s e^{-i\delta t},
\tag{9}
$$

$$
\langle \delta\ddot{Q}_a \rangle + \gamma_{m1}\langle \delta\dot{Q}_a \rangle + \omega_{m1}'^2\langle \delta Q_a \rangle + V'\langle \delta Q_b \rangle = 2g\omega_{m1}(c_s^*\langle \delta c \rangle + c_s\langle \delta c^\dagger \rangle + \langle \delta c^\dagger \rangle\langle \delta c \rangle),
\tag{10}
$$

$$
\langle \delta\ddot{Q}_b \rangle + \gamma_{m2}\langle \delta\dot{Q}_b \rangle + \omega_{m2}'^2\langle \delta Q_b \rangle + V'\langle \delta Q_a \rangle = 2gV(c_s^*\langle \delta c \rangle + c_s\langle \delta c^\dagger \rangle + \langle \delta c^\dagger \rangle\langle \delta c \rangle).
\tag{11}
$$

In order to solve Equations (9)–(11), we use the ansatz as [68]

$$
\langle \delta O \rangle = O_{1+}e^{-i\delta t} + O_{1-}e^{i\delta t} + O_{2+}e^{-i2\delta t} + O_{2-}e^{i2\delta t},
\tag{12}
$$

and substituting Equation (12) into Equations (9)–(11), we can obtian two group equations with ignoring the terms higher than the second-order. The first group describe the first-order sideband as following

$$
\begin{aligned}
(i\Delta + \kappa - i\delta)c_{1+} &= igc_s Q_{a1+} + \sqrt{\kappa_{ex}}\varepsilon_s, \\
(i\Delta + \kappa + i\delta)c_{1-} &= igc_s Q_{a1-}, \\
Q_{a1+} + V_1 Q_{b1+} &= 2g\chi(\delta)(c_s^* c_{1+} + c_s c_{1-}^*), \\
Q_{b1+} + V_3 Q_{a1+} &= 2g\lambda(\delta)(c_s^* c_{1+} + c_s c_{1-}^*).
\end{aligned}
\tag{13}
$$

where $V_1 = V'/(\omega_{m1}'^2 - i\gamma_{m1}\delta - \delta^2)$, $V_3 = V'/(\omega_{m2}'^2 - i\gamma_{m2}\delta - \delta^2)$, $\chi(\delta) = \omega_{m1}/(\omega_{m1}'^2 - i\gamma_{m1}\delta - \delta^2)$, and $\lambda(\delta) = V/(\omega_{m2}'^2 - i\gamma_{m2}\delta - \delta^2)$. Solving the equations, we can obtain

$$c_{1+} = \frac{(\Lambda_2^* + ig\lambda\Pi_1|c_s|^2)\sqrt{\kappa_{ex}}\varepsilon_s}{\Lambda_1(\Lambda_2^* + ig\lambda\Pi_1|c_s|^2) - ig\Lambda_2^*\Pi_1|c_s|^2}, \tag{14}$$

$$c_{1-}^* = \frac{-igc_s^{*2}\Pi_1\Lambda_2^*\sqrt{\kappa_{ex}}\varepsilon_s}{\Lambda_2^*[\Lambda_1(\Lambda_2^* + ig\lambda\Pi_1|c_s|^2) - ig\Lambda_2^*\Pi_1|c_s|^2]}, \tag{15}$$

$$Q_{a1+} = \frac{\Pi_1\Lambda_2^*c_s^*\sqrt{\kappa_{ex}}\varepsilon_s}{\Lambda_1(\Lambda_2^* + ig\lambda\Pi_1|c_s|^2) - ig\Lambda_2^*\Pi_1|c_s|^2}, \tag{16}$$

where $\Lambda_1 = i(\Delta - \delta) + \kappa$, $\Lambda_2 = i(\Delta + \delta) + \kappa$, and $\Pi_1 = 2g(\chi(\delta) - V_1\lambda(\delta))/(1 - V_1V_3)$.

The second group gives the SSG progress as

$$(i\Delta + \kappa - 2i\delta)c_{2-} = igc_sQ_{a2+} + igc_{1+}Q_{a1+},$$
$$(i\Delta + \kappa + 2i\delta)c_{2+} = igc_sQ_{a2-} + igc_{1-}Q_{a1-},$$
$$Q_{a2+} + V_2Q_{b2+} = 2g\chi(2\delta)(c_s^*a_{2+} + c_sc_{2-}^* + c_{1+}c_{1-}^*).$$
$$Q_{b2+} + V_4Q_{a2+} = 2g\lambda(2\delta)(c_s^*a_{2+} + c_sc_{2-}^* + c_{1+}c_{1-}^*). \tag{17}$$

where $V_2 = V'/(\omega_{m1}'^2 - 2i\gamma_{m1}\delta - 4\delta^2)$, $V_4 = V'/(\omega_{m2}'^2 - 2i\gamma_{m2}\delta - 4\delta^2)$, $\chi(2\delta) = \omega_{m1}/(\omega_{m1}'^2 - 2i\gamma_{m1}\delta - 4\delta^2)$, and $\lambda(2\delta) = V/(\omega_{m2}'^2 - 2i\gamma_{m2}\delta - 4\delta^2)$. Solving the equations, we can obtain

$$c_{2+} = \frac{g^2\Pi_3c_s^2c_{1-}^*Q_{a1+} + ig\Pi_3\Lambda_4^*c_sc_{1+}c_{1-}^* + igc_{1+}Q_{a1+}(\Lambda_4^* + ig\Pi_3|c_s|^2) + igc_{1+}Q_{a1+}}{\Lambda_3(\Lambda_4^* + ig\Pi_3|c_s|^2) - ig\Pi_3|c_s|^2\Lambda_4^*}, \tag{18}$$

where $\Lambda_3 = i(\Delta - 2\delta) + \kappa$, $\Lambda_4 = i(\Delta + 2\delta) + \kappa$, and $\Pi_3 = 2g(\chi(2\delta) - V_2\lambda(2\delta))/(1 - V_2V_4)$.

According to the optical cavity input and output theory [69] $c_{out}(t) = c_{in}(t) - \sqrt{2\kappa}c(t)$, we then reach the following relation

$$
\begin{aligned}
\langle c_{out}(t)\rangle &= (\varepsilon_p - \sqrt{\kappa_{ex}}c_0)e^{-i\omega_p t} + (\varepsilon_s - \sqrt{\kappa_{ex}}c_{1+})e^{-i\omega_s t} \\
&\quad - \sqrt{\kappa_{ex}}c_{1-}e^{-i(2\omega_p-\omega_s)t} - \sqrt{\kappa_{ex}}c_{2+}e^{-i(2\omega_s-\omega_p)t} - \sqrt{\kappa_{ex}}c_{2-}e^{-i(3\omega_p-2\omega_s)t}
\end{aligned} \tag{19}
$$

where the first term is the output field with the frequency $\omega_p$, the second one indicates output field with the frequency $\omega_s$, and the third term denotes FWM process with the requency $2\omega_p - \omega_s$. The forth and fifth terms shows the SSG process where $2\omega_s - \omega_p$ (i.e., $\omega_p + 2\delta$) is the second-order upper sideband and $3\omega_p - 2\omega_s$ (i.e., $\omega_p - 2\delta$) is the second-order lower sideband. We introduce a dimensionless FWM intensity to study the FWM process [70]

$$FWM = \left|\frac{\sqrt{\kappa_{ex}}c_{1-}}{\varepsilon_s}\right|^2. \tag{20}$$

In order to research the second-order process conveniently, we use a dimensionless efficiency of SSG as [54–57,62,64]

$$\eta = \left|\frac{-\sqrt{\kappa_{ex}}c_{2+}}{\varepsilon_s}\right|. \tag{21}$$

The parameter values used in the paper [71] are: $\lambda = 1064$ nm, $L = 25$ mm, $\omega_{m1} = \omega_{m2} = 2\pi \times 947$ kHz, $Q_1 = \omega_{m1}/\gamma_{m1} = 6700$, $Q_2 = \omega_{m2}/\gamma_{m2} = Q_1$, $m_1 = m_2 = 145$ ng, $\kappa = 2\pi \times 215$ kHz, $P_c = 0.5$ μW, and $\varepsilon_s = 0.05\,\varepsilon_p$.

## 3. Results and Discussion

### 3.1. The FWM Process

The FWM process has been demonstrated in optomechanical systems [50,51,70], which depends on the intracavity photon number $c_s^2 \propto P_c$ in Equation (20). However, in our hybrid optomechanical system, we concentrate on another two parameters, i.e., the coupling strength $V$ and the frequency difference of the two NRs. Figure 2a plots the FWM spectrum as a function of the probe-cavity detuning $\Delta_s = \omega_s - \omega_c$ for several different coupling strength $V$ with the parameters of the pump power $P_c = 0.5$ μW and the same frequencies $\omega_{m1} = \omega_{m2}$ under the condition of pump on-resonance ($\Delta_c = 0$). In the case of $V = 0$, i.e., the typical FP optomechanical system without considering another NR, we find two sharp sideband peaks (the black curve in Figure 2a) in the FWM spectrum accurately locating at the resonator frequency $\Delta_s = \pm\omega_{m1}$, which can be attributed to the quantum interference of the phonon mode and the beat frequency $\delta$ of two optical fields. Then, if the beat frequency $\delta$ is close to the resonator frequency $\omega_{m1}$, the resonator starts to oscillate coherently leading to Stokes ($\omega_S = \omega_p - \omega_m$) scattering of light from the optomechanical system. However, when another NR is taken into consideration ($V \neq 0$), the sideband peaks locates at $\Delta_s = \pm\omega_{m1}$ splits into two peaks, and with increasing $V$ from $V = 0.1\,\omega_{m1}$ to $V = 0.9\,\omega_{m1}$, the width of the splitting in the FWM spectra is broadening at the expense of intensity as shown the color curves in Figure 2a. When we measure the sideband peaks splitting width of the FWM spectrum under $V \neq 0$, we find that the relation between the splitting width and the coupling strength $V$ of the two NRs is linear as shown in Figure 2b, and the inset in Figure 2b gives the FWM spectrum at a fixed coupling strength $V = 0.3\,\omega_{m1}$. The result will give a method the measure the coupling strength $V$ of the two NRs and we will discuss the result in the following.

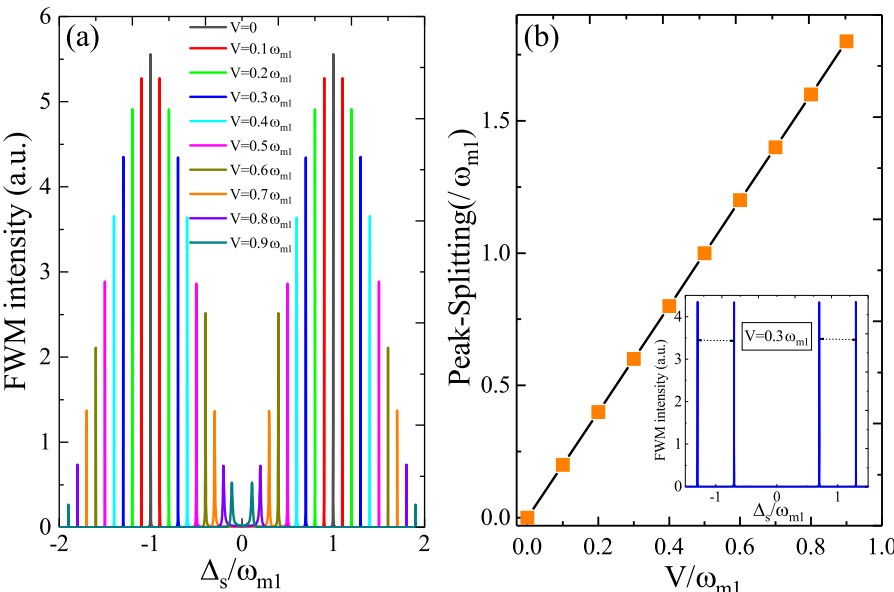

**Figure 2.** (**a**) The FWM spectrum versus the probe-cavity detuning $\Delta_s = \omega_s - \omega_c$ for several different coupling strengths $V$ with the parameters $P_c = 0.5$ μW and $\omega_{m1} = \omega_{m2}$ at $\Delta_c = 0$. (**b**) the peak splitting of the two sideband peaks versus the coupling strength $V$, and the inset in (**b**) give the FWM spectrum at a fixed $V = 0.3\,\omega_{m1}$

Switching the condition to pump off-resonance, i.e., $\Delta_c = \omega_{m1}$, we investigate the FWM process under different parametric regimes. In Figure 3a, we plot the FWM spectra for five different coupling strengths $V$ at $P_c = 5$ μW and $\omega_{m1} = \omega_{m2}$ in the case of red sideband ($\Delta_c = \omega_{m1}$). We can see that there are four sharp sideband peaks in the FWM spectra, which presents mirror symmetry for $\Delta_s = 0$, and the splitting of both sides of the peaks is broadening with enhanced intensity of the FWM spectra for increasing the coupling strength $V$ from

$V = 0.1\,\omega_{m1}$ to $V = 0.9\,\omega_{m1}$. In addition, the location of the sideband peaks is related to the coupling strength $V$. We take $V = 0.3\,\omega_{m1}$ as an example as shown the left inset in Figure 3a, we see that the left two peaks locate at $-1.3\,\omega_{m1}$ and $-0.7\omega_{m1}$, i.e., $-\omega_{m1} \mp V$; while the right two peaks locate at $0.7\,\omega_{m1}$ and $1.3\,\omega_{m1}$, i.e., $\omega_{m1} \pm V$. When measuring the width of the splitting of both of the two sideband peaks in the FWM spectra, we find the width of the splitting is related to the coupling strength $V$ as shown the right inset in Figure 3a, which plots the splitting of both of the two sideband peaks versus the coupling strength $V$. Obviously, the splitting width relies linearly on the coupling strength $V$ and reaches to 0 in the absence of the coupling, which presents an effective means to determine the coupling strength $V$ of the two NRs. Thus, we can measure the coupling strength $V$ of the two NRs via only simply measuring the splitting distance of two sideband peaks in the FWM spectrum. Moreover, we also study the FWM spectrum for different resonator frequencies at a fixed coupling strength $V$. In Figure 3b, we show the FWM spectra as a function of $\Delta_s$ for several different resonator frequencies with the parameters of $P_c = 5\,\mu W$ and $V = 0.3\,\omega_{m1}$ in the case of $\Delta_c = \omega_{m1}$. We find that the peaks in the left part is left-shift and the peaks in the right part is right-shift with increasing the frequency $\omega_{m2}$ from $\omega_{m2} = 0.2\,\omega_{m1}$ to $\omega_{m2} = 2.0\,\omega_{m1}$, and both the intensities of two sideband peaks experience the process of enhancement to decrease. As in Equations (9)–(11), where $\omega_{m1}'^2 = \omega_{m1}^2 + V^2$ and $\omega_{m2}'^2 = \omega_{m2}^2 + V^2$, i.e., the effective frequencies of the two NRs are modulated by the coupling strength $V$ of the two NRs. In Figure 3b, we set a fixed coupling strength $V = 0.3\,\omega_{m1}$, if the two NRs are identical resonators, i.e., the two NRs own the same frequencies $\omega_{m1} = \omega_{m2}$, which is demonstrated in Figure 3a and the splitting width of the sideband peaks in the FWM spectrum is proportional to the coupling strength $V$. In the case of $\omega_{m1} > \omega_{m2}$, i.e., $\omega_{m1}' > \omega_{m2}'$, the FWM spectra is squeezed and both the two sideband peaks move to $\Delta_s = 0$. In the case of $\omega_{m1} < \omega_{m2}$, i.e., $\omega_{m1}' < \omega_{m2}'$, the FWM spectra is expanded and the sideband peaks move to both sides. Compared with FWM process in the optomechanical system only including one mechanical mode and one optical mode [72,73], the intensity of the FWM in our hybrid system is enhanced significantly modulated by another NR.

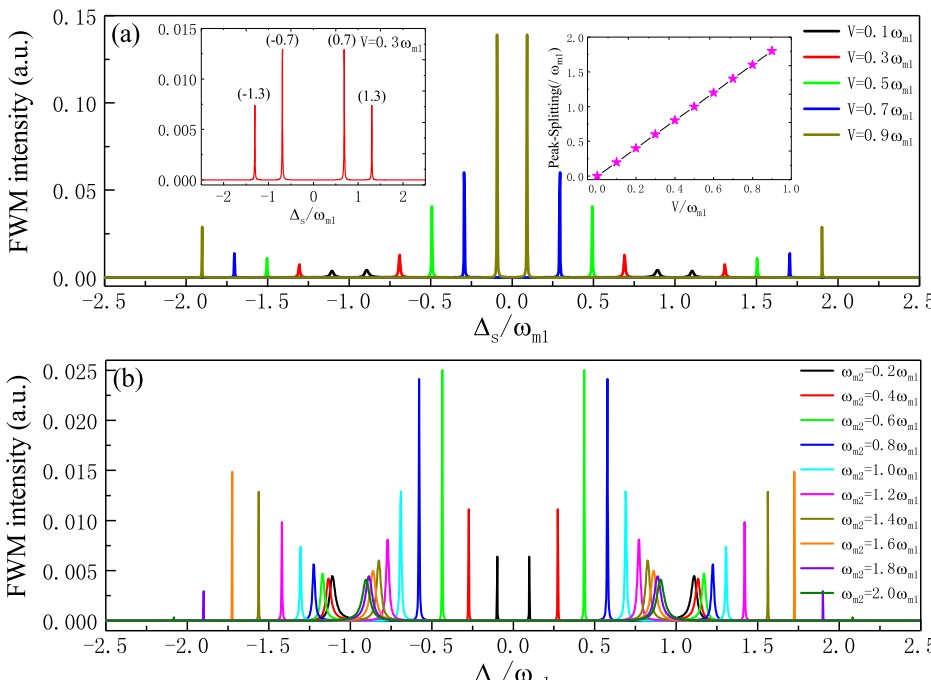

**Figure 3.** (**a**) The FWM spectra for five different coupling strengths $V$ at $P_c = 5\,\mu W$ and $\omega_{m1} = \omega_{m2}$ in the case of $\Delta_c = \omega_{m1}$, and the left inset is $V = 0.3\omega_{m1}$, the right inset plots the splitting of the two sideband peaks as a function of the coupling strength $V$. (**b**) The FWM spectra as a function of $\Delta_s$ for several different resonator frequencies with the parameters $P_c = 5\,\mu W$ and $V = 0.3\,\omega_{m1}$ in the case of $\Delta_c = \omega_{m1}$.

### 3.2. The SSG Process

On the other hand, the SSG is also a distinguished nonlinear phenomenon in optomechanical systems, and Equation (21) gives the dimensionless efficiency of the SSG process. As we know, if the optomechanical system is driven by a two-tone field, the output field with frequency $2\omega_s - \omega_p$ (i.e., $\omega_p + 2\delta$) is the second-order upper sideband and the output field with frequency $3\omega_p - 2\omega_s$ (i.e., $\omega_p - 2\delta$) is the second-order lower sideband. Here, we concentrate on the second-order upper sideband ($\omega_p + 2\delta$) and study the coupling strength $V$ and the frequency difference of the two NRs the affect the SSG process for different detuning. In Figure 4, we show the efficiency $\eta$ of SSG as a function of the normalized probe detuning $\Delta_s/\omega_{m1}$ for several different coupling strengths $V$ with the parameters of the pump power $P_c = 1.5\ \mu W$ and the same frequencies $\omega_{m1} = \omega_{m2}$ under the condition of pump on-resonance $\Delta_c = 0$. If the system is the typical FP optomechanical system without considering another NR, i.e., $V = 0$, as shown the black curve in Figure 4, it is clear that four sideband peaks emerge in the efficiency $\eta$ of SSG, where two principal sideband peaks locate at $\Delta_s = \pm\omega_{m1}$ and another two secondary sideband peaks locate at $\Delta_s \approx \pm 0.5\ \omega_{m1}$ with a small shift. We accurately identify the location of the principal sideband peaks at $\pm 1.0\ \omega_{m1}$ and secondary sideband peaks at $-0.46\ \omega_{m1}$ and $0.53\ \omega_{m1}$, respectively, as shown the black curve in Figure 4. When another NR is considered ($V \neq 0$), we find that both the principal sideband peaks locating at $\pm 1.0\ \omega_{m1}$ and the secondary sideband peaks around $\pm 0.5\ \omega_{m1}$ split into two peaks, and then eight sideband peaks appear in efficiency $\eta$ of SSG. Here, we take $V = 0.1\ \omega_{m1}$ as an example as shown the inset in Figure 4. The splitting of principal sideband peaks locate at $-1.1\ \omega_{m1}$ and $-0.9\ \omega_{m1}$ (i.e., $-\omega_{m1} \mp V$), and $0.9\ \omega_{m1}$ and $1.1\ \omega_{m1}$ (i.e., $\omega_{m1} \pm V$) with the width of splitting of $2\ V$. The splitting of secondary sideband peaks are located at $(-0.53, -0.43)\ \omega_{m1}$ and $(0.46, 0.56)\ \omega_{m1}$, i.e., $\pm\omega_{m1} - 0.07\ V$ and $\pm\omega_{m1} + 0.03\ V$, respectively, with the width of splitting of $V$.

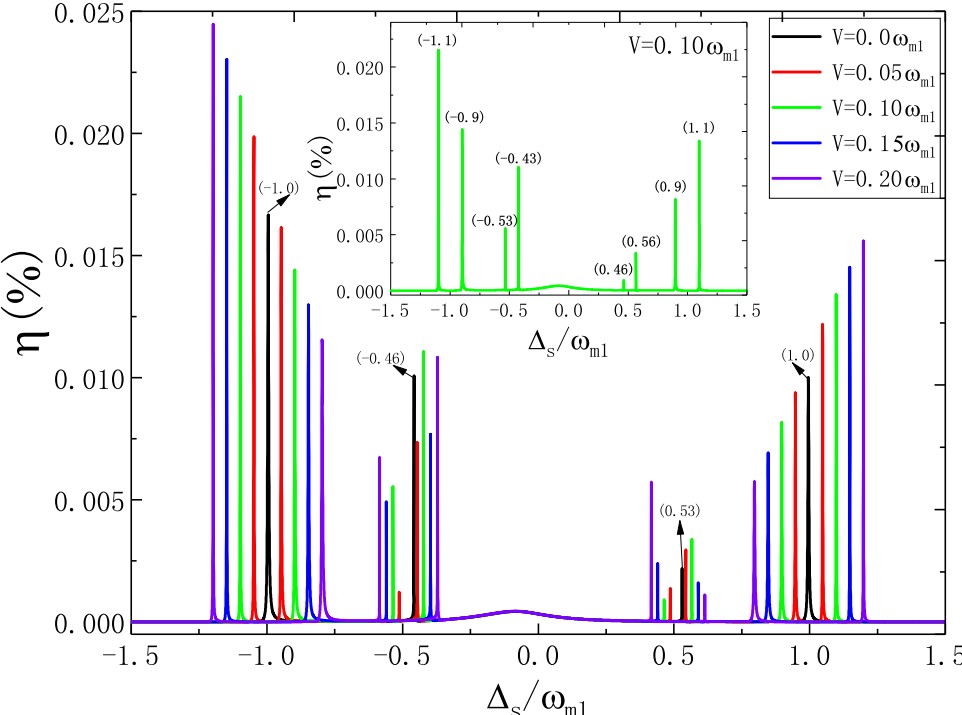

**Figure 4.** The efficiency $\eta$ of SSG versus $\Delta_s/\omega_{m1}$ for several different coupling strengths $V$ with the parameters $P_c = 1.5\ \mu W$ and $\omega_{m1} = \omega_{m2}$ under the condition of $\Delta_c = 0$, and the inset gives the condition of $V = 0.1\ \omega_{m1}$.

Moreover, we also investigate the different resonator frequencies of the two NRs that influence the SSG process as shown in Figure 5. Figure 5b gives the efficiency $\eta$ of SSG

versus $\Delta_s/\omega_{m1}$ for the same frequencies of the two NRs at the parameters of $P_c = 1.5\ \mu W$ and $V = 0.1\ \omega_{m1}$ under the pump on-resonance $\Delta_c = 0$, which manifests four principal sideband peaks locating at $\pm\omega_{m1} \pm V$ and four secondary sideband peaks locating around $\Delta_s \approx \pm 0.5\ \omega_{m1}$, and the precise locations are marked in Figure 5b. If $\omega_{m2} < \omega_{m1}$ as shown in Figure 5a, we find the spectrum of the SSG is squeezed, while if $\omega_{m2} > \omega_{m1}$ as shown in Figure 5c, we can see that the spectrum of the SSG is stretched, and the accurate locations are identified in Figure 5a,c, respectively.

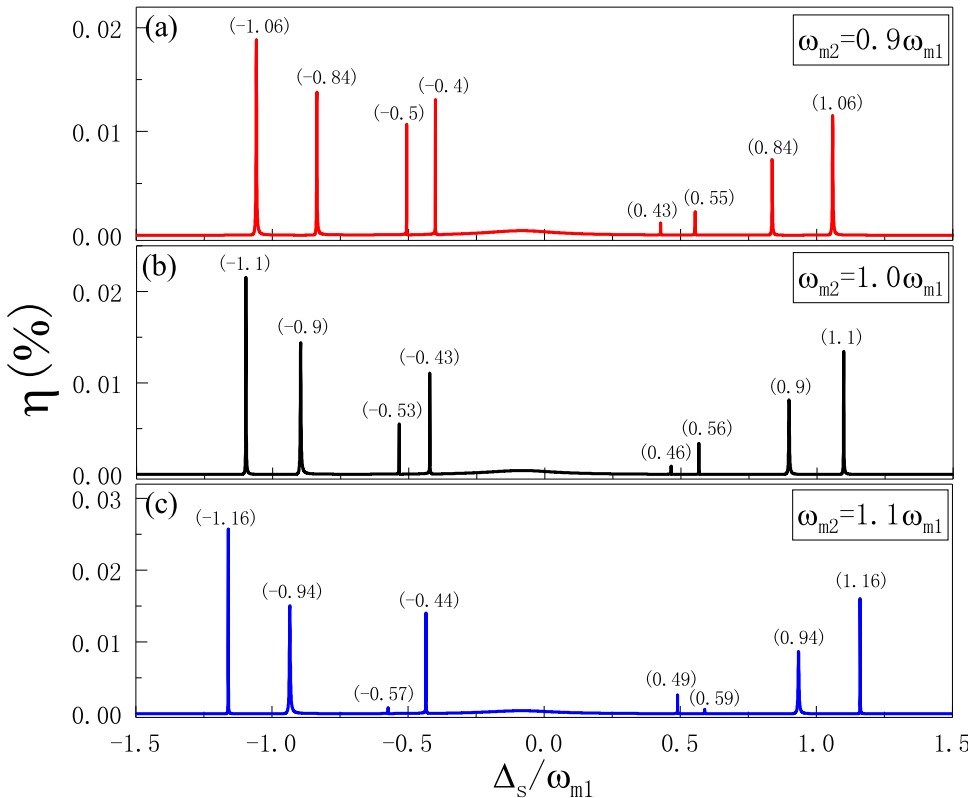

**Figure 5.** (**a**) The efficiency $\eta$ of SSG as a function of $\Delta_s/\omega_{m1}$ for $\omega_{m2} = 0.9\omega_{m1}$. (**b**) The efficiency $\eta$ of SSG as a function of $\Delta_s/\omega_{m1}$ for $\omega_{m2} = 1.0\omega_{m1}$. (**c**) The efficiency $\eta$ of SSG as a function of $\Delta_s/\omega_{m1}$ for $\omega_{m2} = 1.1\omega_{m1}$. The other parameters are $P_c = 1.5\ \mu W$, $V = 0.1\omega_{m1}$, and $\Delta_c = 0$.

Then, we further research the SSG for several different parametric regimes in the condition of $\Delta_c = \omega_{m1}$. Figure 6a presents the efficiency $\eta$ of SSG versus $\Delta_s/\omega_{m1}$ with four different coupling strengths $V$ at $P_c = 2.0\ \mu W$ and the same frequencies $\omega_{m1} = \omega_{m2}$. When $V = 0$ as shown the black curve in Figure 6a, i.e., without considering another NR, we find that not only a second-order sharp single peak locating at $\Delta_s = 0.5\ \omega_{m1}$ but also a second-order OMIT sideband locating at $\Delta_s = 1.0\ \omega_{m1}$ with symmetrical splitting appear in the SSG spectrum. Once another NR is taken into consideration, i.e., $V \neq 0$, we can obtain two significant phenomena in the SSG spectra: (a) the single peak locating at $\Delta_s = 0.5\ \omega_{m1}$ splits into two peaks; (b) the second-order OMIT sideband locating at $\Delta_s = 1.0\ \omega_{m1}$ splits into double second-order OMIT sideband with asymmetrical splitting. The phenomenon originates from the coupling between $NR_1$ and $NR_2$ which not only adds a fourth level, as shown in Figure 1b, but also breaks down the symmetrical second-order OMIT sideband due to quantum interference and, therefore, induces sharp bright resonance within the second-order OMIT sideband. It is obvious that due to the interaction of the two NRs via Coulomb interaction, the coupled number states of the photon and phonons are induced. Then, the symmetrical second-order OMIT sideband is split into two asymmetrical double second-order OMIT sideband. Here, we take $V = 0.1\ \omega_{m1}$ as an example as shown in the inset of Figure 6a, we find the two splitting peaks located at $0.45\ \omega_{m1}$ and $0.55\ \omega_{m1}$ with

the width $V$ of the splitting, i.e., the two sharp peaks locate at $\omega_{m1} \pm V/2$. While another double asymmetrical second-order OMIT sidebands locate at $0.9\omega_{m1}$ and $1.1\ \omega_{m1}$ with the width $2V$ of the splitting, i.e., the double second-order OMIT sidebands locate at $\omega_{m1} \pm V$. With further increasing the coupling strengths $V$ of the two NRs, both the splitting of the two sharp peaks and the double second-order OMIT sidebands are enhanced. Moreover, we also investigate the frequency difference of the two NRs that affect the efficiency $\eta$ of SSG. In Figure 6b, we show the efficiency $\eta$ of SSG as a function of $\Delta_s/\omega_{m1}$ for three different frequencies at $P_c = 2.0$ μW and $V = 0.1\ \omega_{m1}$. It is obvious that the SSG spectra show two splitting peaks around $\Delta_s = 0.5\ \omega_{m1}$ and double asymmetrical second-order OMIT sidebands around $\Delta_s = 1.0\ \omega_{m1}$, and if $\omega_{m2} < \omega_{m1}$, the SSG spectrum shifts to the left, if $\omega_{m2} > \omega_{m1}$, the SSG spectrum shifts to the right. Therefore, the double second-order OMIT sidebands can be controlled with manipulating the coupling strength $V$ and the frequency difference of the two resonators.

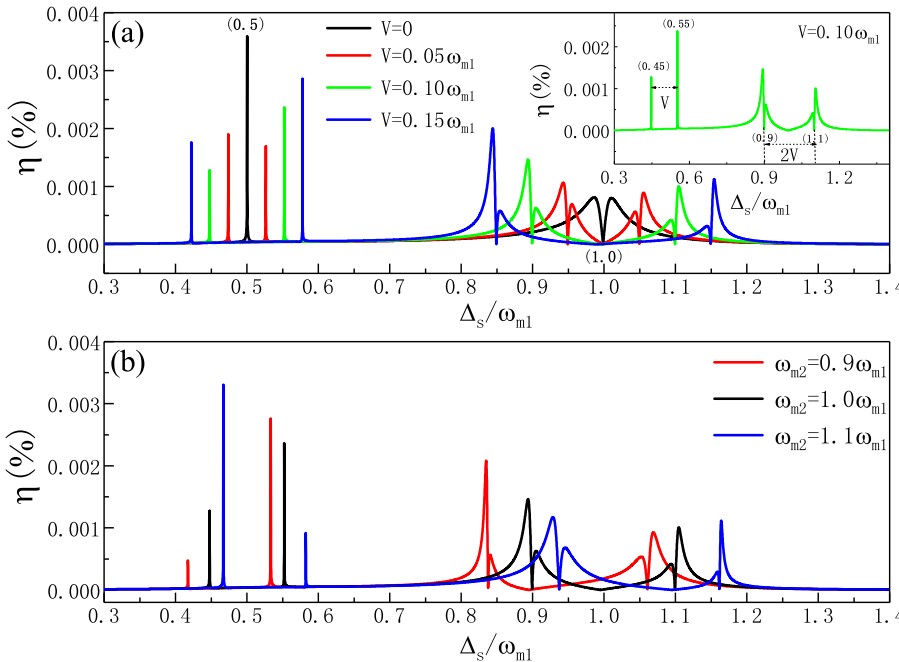

**Figure 6.** (**a**) The efficiency $\eta$ of SSG four different coupling strengths $V$ with the parameters of $P_c = 2.0$ μW and $\omega_{m1} = \omega_{m2}$ at $\Delta_c = \omega_{m1}$, and the inset gives $V = 0.1\ \omega_{m1}$ as an example. (**b**) The efficiency $\eta$ of SSG as a function of $\Delta_s/\omega_{m1}$ for three different frequencies at $P_c = 2.0$ μW and $V = 0.1\ \omega_{m1}$.

Compared with the SSG in the typical FP optomechanical system without considering another NR, there are several advantages of the SSG in our hybrid optomechanical system. In the condition of $\Delta_c = 0$, if only one NR in the system ($V = 0$), the SSG spectrum presents the sharp principal sideband peaks at $\pm 1.0\ \omega_{m1}$ and secondary sideband peaks around $\pm 0.5\ \omega_{m1}$, while if another NR is considered ($V \neq 0$) in the case of identical NRs with the same frequencies ($\omega_{m2} = \omega_{m1}$), both the sharp principal sideband peaks and secondary sideband peaks are split into two peaks, and the splitting width of the principal sideband peaks is linear with respect to the coupling strength $V$ of the two NRs, which indicates a method to precisely determine the coupling strength $V$ of the two NRs. In addition, if the two NRs are different with different frequencies ($\omega_{m2} \neq \omega_{m1}$), the location of sharp sideband peaks in the SSG spectra are tunable. On the other hand, in the condition of pump off-resonance, i.e., $\Delta_c = \omega_{m1}$, the SSG shows a second-order OMIT sideband [64] at $V = 0$, while if another NR is taken into consideration ($V \neq 0$), the SSG displays double second-order OMIT sidebands which is very different from the case of $V = 0$. Therefore, the SSG process can be tunable via controlling the coupling of the two NRs.



## 4. Conclusions

We theoretically demonstrated the FWM and SSG in a hybrid COM system which is driven by a two-tone field for different detuning conditions, where an optomechanical resonator is coupled to another NR via Coulomb interaction. We first studied the FWM under the condition of pump on-resonance and pump off-resonance, when the coupling strength $V$ of the two NRs is considered, the FWM presents four modes splitting, which gives a method to determine the coupling strength $V$. In addition, the frequency difference of the two NRs also alter the FWM process. On the other hand, another nonlinear phenomena of the SSG process has also demonstrated both in pump on-resonance and red sideband with controlling the parameters of the coupling strength $V$ and the frequency difference of the two NRs. Moreover, the SSG is sensitive to the detuning, which displays the double asymmetrical second-order OMIT sidebands via controlling $V$ and frequencies of the resonators, which may indicate a further insight of nonlinear optomechanical phenomena and may find important applications fot manipulating light propagation and quantum communications based on the hybrid optomechanical system.

**Funding:** This research was funded by National Natural Science Foundation of China (Nos:11647001 and 11804004), Project funded by China Postdoctoral Science Foundation (No:2020M681973) and Anhui Provincial Natural Science Foundation (No:1708085QA11).

**Institutional Review Board Statement:** Not applicable.

**Informed Consent Statement:** Not applicable.

**Data Availability Statement:** Not applicable.

**Acknowledgments:** Huajun Chen is supported by the National Natural Science Foundation of China (Nos:11647001 and 11804004), Project funded by China Postdoctoral Science Foundation (No:2020M681973) and Anhui Provincial Natural Science Foundation (No:1708085QA11).

**Conflicts of Interest:** The authors declare that they have no known competing financial interests or personal relationships that could have appeared to influence the work reported in this paper.

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
