# Peer review of "Robust Four-Wave Mixing and Double Second-Order Optomechanically Induced Transparency Sideband in a Hybrid Optomechanical System"

_photonics, doi:10.3390/photonics8070234_

Round 1

Reviewer 1 Report

The results in this work are interesting and relevant for a journal like the Photonics. Therefore, I recommend it for publication in this journal after a minor revision. The following questions should be answered. 1. In figure 2, the peak of the FWM is split due to the coupling between the two resonators. Is the distance between the two splitting peaks dependent on the coupling strength like that in figure 3(a)? Please explain their relations. 2. In figure 3(b), while changing the resonator frequency $\omega_{m2}$, the FWM peaks have a noticeable shift. Please give a reasonable explanation.

Reviewer 2 Report

In this manuscript Chen explores the nonlinear optical interactions in a cavity optomechanical system. The four wave mixing and second order sideband generation functions have previously been studied for optomechanical systems with a single resonator. Here, a new scenario involving two coupled resonators is explored in the same context. The author finds and outlines a rich set of sidebands generated from the process. The results and theoretical model are clearly outlined.

This work expands the possibilities for nonlinear optics with cavity optomechanical systems. This could form the basis for new and more varied experiments. The goal of these experiments beyond more conventional nonlinear optics is not completely clear, but the work is a valuable expansion of existing literature.

I have a few remaining questions and comments which I outline below. If these can be addressed within the manuscript, I am happy to recommend publication within Photonics.

  1. The methods used to generate the results plots are completely absent from the manuscript. Were the presented equations solved analytically or numerically simulated? This process should be outlined.
  2. The motivation for this work is still a little unclear. In the conclusion it states that experimentalists could use the location of these sidebands to determine the coupling strength V between the two resonators. This seems complicated compared to more direct methods such as exciting one resonator resonantly and observing the exchange of the mechanical energy directly in the optical signal. Are there perhaps other applications more directly related to the emitted sidebands?
  3. In the manuscript the pump power Pc is given, but the probe power Ps is not given. Is this constant? What values are used for each section of the results?
  4. For the four wave mixing the results are always given in arbitrary units. Are these actually arbitrary units or do these correspond to equation 20? If not it would be helpful to have some indication of the scale, at least in the text.
  5. The experimental parameters from Ref. 68 are chosen as the basis for this work. What was the reason for choosing this particular experiment? Is it particularly suitable for nonlinear optomechanics?
  6. The first line states that cavity optomechanics is a milestone from 2006. Investigations into cavity optomechanical experiments and theory were already started long before (see for example work from the early years of gravitational wave detection research). Furthermore, as is noted, pushing cavity optomechanics to the quantum regime happened some years later. I believe the date should be removed.
  7. There are a few small typos in the work:
    • Line 94 attributes->can be attributed
    • Line 99 locating->locates

Reviewer 3 Report

Journal: Photonics

Manuscript ID: photonics-1244959

Title: Robust four-wave mixing and double second-order optomechanically induced transparency sideband in a hybrid optomechanical system

Authors: Hua-Jun Chen*

In the submitted manuscript (MS) titled “Robust four-wave mixing and double second-order optomechanically induced transparency sideband in a hybrid optomechanical system”, the author theoretically investigates the four-wave mixing (FWM) and second-order sideband generation (SSG) in a hybrid optomechanical system in which an optomechanical resonator is coupled to the other nanomechanical resonator via Coulomb interaction. He/She shows that under what he/she called the condition of resonance and off-resonance, the FWM and SSG can be obtained and controlled by adjusting the system’s parameters.

From my point of view, the research on the issue to reveal the nonlinear optical phenomena resulting from the intrinsic nonlinearity arising from optomechanical interaction in various hybrid optomechanical systems is an interesting subject, as this MS focusing on. However, I think that in the MS the author exaggerates its impact and innovation, the logic of writing and discussion is a bit confusing, and it probably won’t draw much interest from the community of optomechanics and this MS is not suitable for publication in Photonics. Below I summarize my main objections.

  1. In the MS, the author obtains the Hamiltonian (1) by referring to the papers [63-65]. The fifth term is used to describe the interaction of two charged NRs, which is different from that in Refs. [63-65] in which the interaction of two charged NRs is described by using the corresponding position operators. So, how did the author obtain the fifth term and prove it is effective? In particular, in the process of the following calculation, the author turns to use position operators to describe two charged NRs with Qa,b=b1,2^+ + b1,2 which is different from the general relational expression between position operators and number operators. These inconsistencies confuse readers.

  1. To obtain FWM and SSG in a generic linearly coupled optomechanical system, the one-phonon resonance condition needs, in general, to be met, i.e., the detuning between the cavity and the pump driving laser is always selected as Δc≈ωm1, due to the fact that the underlying physical process in the linearly coupled optomechanical system implies one-phonon processes. And in this MS, the author chooses the detuning between the cavity field and the pump driving laser as Δc=0 as the so-called resonance condition. So, what is the physical reason for this choice?

  1. I have noticed that the author plots the FWM and SSG spectra as a function of the so-called probe-cavity detuning Δs, not the probe-pump detuning δ, which is so different from that in the previous papers. So, what is the reason for this choice? In fact, the definition of the so-called probe-cavity detuning Δs is not given before Section 3 and the parameter Δs does not appear in any expressions from Eqs. (1)-(21).

  1. In lines 132-134, the author states that “…, the output fields with frequencies  Δp±2nδ can emerge, where n is an integer representing the order of the sidebands, …”. In fact,  Δp±2nδ should be Δp±2δ.

  1. There are many discussions about the locations of peaks in the output spectra in the MS. However, I think that these discussions confuse the readers. Here I just give one example. In lines 111-112, the author states that “…, we see that the left two peaks locate at -1.3ωm1 and -0.7ωm1, i.e., -Δs-+V; …” According to this statement and Fig.3(a), one can obtain the equations Δs=-1.3ωm1=-Δs-V and Δs=-0.7ωm1=-Δs+V. Then the value of V obtained from the two equations is not the same. So, how does the author explain this result?

  1. There is one word “Robust” in the title of the MS. How do the findings reflect the “robust”? After all, the word “Robust” does not appear in the text of the MS.

  1. Figure 1(b), that is, the energy level diagram of the system is not discussed in the text of the MS.

  1. The last paragraph in Section 1 is more like a conclusion.

  1. The author should also set different linetypes to distinguish curves in the figures.
  2. In the MS, there are many incorrect language expressions. Thus, I think the quality of English in the MS is such that the manuscript cannot be published in its present form.

I conclude that the present work does not meet the criteria of publication in Photonics. I suggest the author writes and checks the MS carefully, adds more analysis, and shows the innovative points clearly if any. In view of this, I cannot recommend publication in Photonics.

Round 2

Reviewer 3 Report

The author has successfully addressed most of my comments. And I am satisfied with most of the answers by the author overall. Although I think this revised manuscript is not the best, now I would recommend it for publication in Photonics.